# The doctor knows or the evidence shows: An online survey experiment testing the effects of source trust, pro-vaccine evidence, and dual-processing in expert messages recommending child COVID-19 vaccination to parents

Ava Irysa Kikut *

Annenberg School for Communication, University of Pennsylvania, Philadelphia, Pennsylvania, United States of America

* ava.kikut@asc.upenn.edu

## Abstract

Increasing child vaccination rates is a critical step toward mitigating the spread of COVID-19. Both distrust in expert sources and concern about the safety and efficacy of vaccines may contribute to parent vaccine hesitancy. The present study is the first to test the effectiveness of building trust and providing evidence supporting child COVID-19 vaccines in recommendation messages for parents. Based on dual-processing theories, emphasis on source trustworthiness and pro-vaccine evidence may each be particularly effective when the other is not present. It was hypothesized that these two approaches would have main and interaction effects on perceived message effectiveness and pro-vaccine beliefs. A between-subjects 2 (trust-building appeal vs. no trust-building appeal) X 2 (pro-vaccine evidence vs. no pro-vaccine evidence) online survey experiment was conducted in December 2021 and January 2022 with United States parents/guardians of children <18 years old ($n$ = 401). As hypothesized, trust and pro-vaccine evidence each had significant simple main effects on both outcomes. Analysis of variance showed a significant negative interaction effect of trust and pro-vaccine evidence on perceived message effectiveness [$F_{(3, 394)}$ = 6.47; $\eta^2$ = 0.02, p = 0.002; 95% CI (0.01, 0.11)], supporting the dual-processing hypothesis. The interaction effect on pro-vaccine beliefs was also negative but not significant [$F_{(3, 394)}$ = 2.69; $\eta^2$ = 0.01; $p$ = 0.102; 95% CI (0.00, 0.03)]. Either highlighting evidence supporting vaccines or building trust in expert sources can influence parent vaccine support. Messages which include strong evidence supporting recommended behaviors may influence recommendation acceptance even among those with lower trust in expert sources and establishing trust may reduce the need to describe available evidence.

**Data Availability Statement:** All relevant data are within the paper and its Supporting information files.

**Funding:** The author received no specific funding for this work.

**Competing interests:** The authors have declared that no competing interests exist.

## Introduction

COVID-19 vaccinations for children have been established as safe, effective, and critical for helping the population reach herd immunity [1]. However, as of December 2021, among United States (U.S.) parents of children between five and eleven years old, one-third did not intend to get their children vaccinated and another third were uncertain [2]. This study was undertaken as the U.S. Food and Drug Administration considered approval of vaccination for children under five-years-old. A critical question for public health then, and still now, is which messaging approaches are effective in addressing parent vaccine hesitancy [3].

The present study focuses on two potential approaches for recommending child COVID-19 vaccines to parents: providing scientific evidence supporting the safety and efficacy of the vaccine and building trust in the intentions of expert sources. Prior research supports both approaches [e.g., 4–6], though they have not been tested experimentally in the context of parent COVID-19 vaccine decisions. Further, this study focuses specifically on the interaction of pro-vaccine evidence and source trustworthiness in recommendation messages, asking whether including evidence supporting vaccines reduces the need for trust and whether building trust reduces the need for pro-vaccine evidence.

On the one hand, these two messaging strategies may complement each other; perceiving a source as trustworthy may increase acceptance of the pro-vaccine evidence the source provides [5, 7–9]. On the other hand, dual-processing theories suggest individuals will be less concerned with source trustworthiness when strong evidence supporting vaccination is provided [10–13]. Previous research has found source trustworthiness and pro-vaccine evidence may both be effective messaging strategies, though few have examined their main and interaction effects in the context of COVID-19 vaccine messages to parents.

### Providing pro-vaccine evidence as a messaging approach

Outside of COVID-19 vaccination, several studies have examined vaccine hesitancy among parents. One of the most common types of concerns among hesitant parents is vaccine safety [14]. For example, parents who are vaccine hesitant may believe vaccines will weaken their children's immune system [15–17] or lead to serious side effects [15, 16, 18]. Another class of concerns center around vaccine efficacy and necessity [14]; vaccine hesitant parents may be more likely to question the effectiveness of vaccines and to prefer their children develop immunity through natural disease rather than vaccination [15, 16, 19].

Specific barriers to child COVID-19 vaccination include parent concerns about side effects, belief that the vaccine was developed too quickly to generate strong safety evidence, and the view that children are not at risk of getting seriously ill from COVID-19 [20]. Factors that have been associated with parent COVID-19 vaccine acceptance include knowledge about vaccines, belief in their safety and effectiveness, high perceived risk of COVID-19, and reliance on health care providers as sources of COVID-19 information [21]. Overall, the proportion of parents who show COVID-19 vaccine hesitance is higher than the proportion showing hesitance for routine child vaccines—calling for more research focused specifically on this issue [21].

While there is limited prior work testing COVID-19 vaccine communication strategies directed toward parents, a few studies have identified approaches for either reducing general COVID-19 vaccine hesitancy [4] or parent hesitancy pertaining to other vaccines [6, 22, 23]. One approach to increasing openness to vaccines is highlighting the scientific evidence supporting their safety and efficacy. A recent survey experiment testing COVID-19 vaccine messaging found positive effects of providing pro-vaccine evidence and information on vaccine acceptance [4], though this study did not focus on parents. Beyond the COVID-19 context,

providing parents with evidence that vaccines benefit their children has been effective in encouraging measles-mumps-rubella vaccination [6], influenza vaccination [22], and human papillomavirus vaccination [23].

## Building trust in the source as a messaging approach

In addition to describing evidence supporting vaccines, another promising approach to increasing vaccine acceptance is building trust [5]. Prior research has found vaccine hesitant parents are more likely to believe health care providers are pressured by vaccine manufacturers to recommend vaccines and are motivated by self-interest [19]. Experts may reduce these biases by demonstrating their motivation to provide recommendations aligning with children's best interests. Recognition of shared values has been shown to build trust in a source's intentions (i.e., source trustworthiness) [24]. Trust in a source can increase attention to messages [25] and strengthen message effects on behavioral beliefs [7, 8]. Specific to COVID-19 vaccine messaging, there is some evidence source trustworthiness increases message effectiveness [9]. In the context of child vaccines, trusting the recommendation source's motivations has been shown to decrease parent hesitancy [26].

## Dual-processing and the potential interaction between approaches

While both pro-vaccine evidence and source trustworthiness may independently increase parent acceptance of vaccine recommendations, one important question is how these approaches complement one another. Dual-processing theories posit that individuals make decisions based on one of two types of processes: heuristic (also called Type 1, automatic, or peripheral processing) and systematic (also called Type 2, deliberate, central processing) [10–13]. While heuristic processing usually involves reliance on cognitive shortcuts to inform decisions, systematic processing involves attention to the quality of message arguments [12, 13]. Trust in a source may serve as a heuristic, reducing complexity when information relevant to the recommended behavior is unavailable and one is unable to engage in systematic processing [12, 13, 27, 28]. That is, trust may allow people to defer to others with whom they share similar values rather than form decisions based on their own knowledge [28]. However, when the information to make reasoned judgments is accessible, source trustworthiness may have a weaker influence on judgment formation [13].

Supporting dual-processing, prior research has found systematic processing of strong arguments can override the influence of trust on message acceptance [29–33]. In the COVID-19 context, some studies have found systematic processing and high COVID-19 specific knowledge were positively associated with accepting COVID-19 pandemic recommendations [34, 35]. However, these studies do not test the implications of dual-processing for communicating vaccine recommendations and whether pro-vaccine evidence decreases reliance on message features which are less directly relevant to the consequences of engaging in the recommended behavior (e.g., source trustworthiness).

One observational study found that believing in the safety and efficacy of vaccines could attenuate the association between trust in public health sources and subsequent vaccination behavior [36]. An experiment testing the interaction effect of vaccine-specific knowledge and source trustworthiness on COVID-19 vaccination outcomes has the potential to strengthen causal claims. Further, prior work has not focused on parent vaccine decisions. The current study contributes to the dual-processing literature by testing the interaction of recommendation-specific information and source trustworthiness in this new context.

### The present study

Building on prior research, this experimental study examines the effects of pro-vaccine evidence, source trustworthiness, and their interaction, in expert messages encouraging parents to vaccinate their children against COVID-19. The present study is not only theoretically important, but practically relevant. As public trust in official health sources and institutions continues to decline [37, 38], it is vital to understand how to communicate effectively to audiences with low trust in experts. Based on dual-processing theories and prior research, offering clear evidence supporting vaccine safety and efficacy may reduce the gap in vaccine acceptance between those who trust and those who distrust expert sources. Yet, individuals may also feel uncertain about the safety and efficacy of newly developed vaccines. When this uncertainty is not addressed in a recommendation message, building trust in expert intentions may be particularly helpful in increasing vaccine acceptance. Drawing from prior research and dual-processing theories, the following was hypothesized (https://aspredicted.org/ZNJ_YB2):

**Hypothesis 1**: Building source trustworthiness will have positive main effects on message effectiveness.

**Hypothesis 2**: Exposure to evidence supporting vaccine safety and efficacy will have positive main effects on message effectiveness.

**Hypothesis 3**: There will be a negative interaction effect of building source trustworthiness and providing evidence supporting vaccine safety and efficacy on message effectiveness, such that each approach will have stronger positive effects when the other is not used.

## Materials and methods

### Sample

An online survey experiment was programmed in Qualtrics^XM and administered through the Forthright Capabilities panel [39]. Forthright is an online research panel that recruits panelists through digital advertising and mail campaigns using address-based probability sampling. Panelists are screened with security checks and asked to provide basic demographic information for a profiling database upon enrolment in the panel. The researcher provides Forthright with information on eligibility criteria and sample size. When a new study is live, Forthright alerts potentially eligible respondents via email, offering information about the survey length and compensation. Once the sample size target is met, or slightly exceeded, the survey is closed to panelists.

Participants were eligible for the current study if they resided in the U.S. and were parents or guardians of at least one child under 18 years of age. In total, 420 Forthright panelists received an invitation to participate in the study (out of a pool of approximately 5000 eligible panelists) through stratified random sampling using nationally representative quotas for gender, age, region, and ethnicity. Of those invited, 401 panelists participated (response rate = 95.5%). Due to budget constraints, the survey experiment was conducted in two phases. An initial sample of 200 participants were recruited to complete the experiment between 12/9-12/11/2021 ($n = 200$). An additional 201 participants were recruited to complete the same experiment between 1/25-1/29/2022 ($n = 201$). The samples were combined. Hypotheses were established prior to the initial round of data collection and remained unchanged. Consent to participate in the panel was obtained from all participants. This study was approved by the University of Pennsylvania's institutional review board.

## Design

The survey procedure is illustrated in Fig 1. After opening the survey on their Forthright dashboard and agreeing to participate, respondents were presented with one of four quotes recommending COVID-19 vaccination. All quotes were attributed to the same fictional physician (Dr. Taylor Clark). To help ensure the whole quote was read, respondents were not shown the button allowing for progression to the next page for the first 30 seconds the quote was displayed. After reading the quote, all participants were asked the same set of questions, including outcome measures (perceived message effectivess and pro-vaccine beliefs) and manipulation check measures (perceived source trustworthiness and whether quote included evidence). The mean response time for the experiment was 3.3 minutes.

Participants were randomly assigned to one of four conditions—*trust cue-only*, *pro-vaccine evidence-only*, *trust cue + pro-vaccine evidence*, or *no trust cue + no pro-vaccine evidence (control)* with a modified quote for each condition. The trust cue, which was intended to increase perceived source trustworthiness, was a description of the source's decision to have their own child vaccinated (e.g., "As a parent. . .I wouldn't recommend this vaccine if I didn't think it was safe for my own family. I got my children vaccinated"). Pro-vaccine evidence focused on scientific research and data supporting the safety and effectiveness of the vaccine for children (e.g., "Clinical trials with thousands of children have shown the vaccine is over 91% effective. Less than 0.01% of children have experienced serious side effects. . .The data is clear."). To ensure the three passages were similar in length (520–527 characters), the combined trust cue + pro-vaccine evidence condition included trimmed-down versions of the trust cue and vaccine evidence. Quotes were constructed based on YouTube videos [40] and opinion pieces [41] with physicians recommending the COVID-19 vaccine for children. All evidence provided was supported by peer-reviewed studies [1] and public health agencies [42, 43].

For all participants, the quote ended with the same sentence ("I recommend the COVID-19 vaccine to all who are eligible."). The control condition quote included only this sentence. Each full quote can be found in supplementary materials (S1A Table in S1 File).

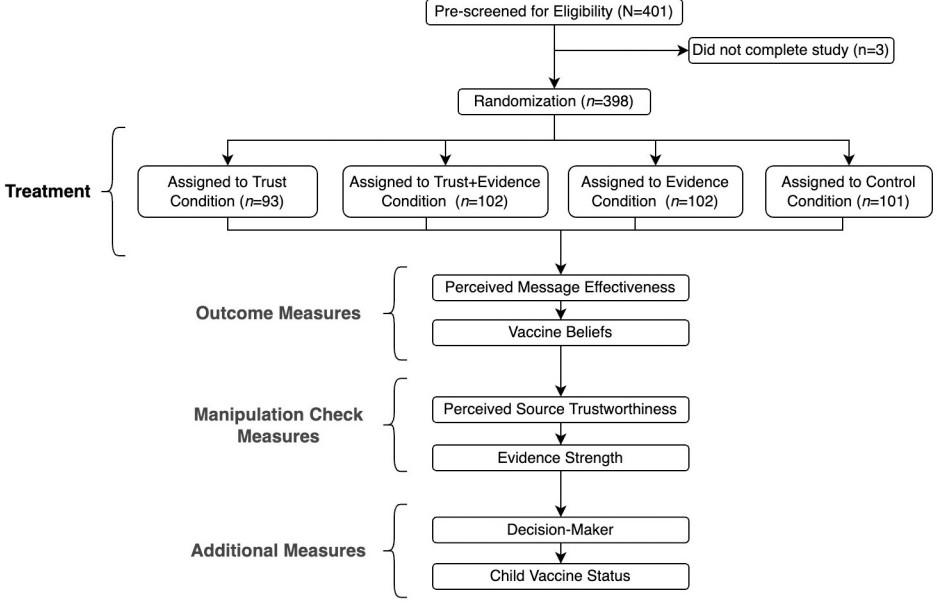

**Fig 1. Flow chart illustrating study procedure.**

## Manipulation checks

A manipulation check measured whether trust cues influenced perceived source trustworthiness as intended. Perceived trustworthiness was measured using a six-item scale (alpha = 0.89) [e.g., *The speaker cares about my child's well-being*; *I trust the speaker to be honest*]. Responses ranged from 1 (low trust in the source) to 6 (high trust in the source). Items were adapted from McCroskey's character scale (18) and the full battery can be found in S1 File. Mean trustworthiness was 5.35 (SD = 1.31) with the trust-cue, compared to 4.64 (SD = 1.65) without the trust cue. An analysis of variance (ANOVA) found significant effects of trust cues on perceived source trustworthiness [$F(3,394) = 22.26$; $\eta^2 = 0.05$, $p < 0.001$; 95% CI (0.02, 0.10)].

A second manipulation check measure assessed whether participants recognized evidence in favor of children receiving the COVID-19 vaccine in the quote they read [*The quote I read for this survey mentioned research that has been done on COVID-19 vaccines with children; The quote I read for this survey gave me lots of evidence in favor of getting my child vaccinated* (1 = No; 2 = Yes)]. These two items (r = 0.52) were combined to create a perceived evidence strength score (0.5 = weak evidence; 2.5 = strong evidence). The mean perceived evidence strength was 1.86 (SD = 0.51) when pro-vaccine evidence was included, compared to 1.28 (SD = 0.67) when evidence was excluded. An ANOVA found perceived evidence strength was significantly higher for participants exposed to evidence [$F(3,394) = 95.00$; $\eta^2 = 0.19$, $p < 0.001$; 95% CI (0.13, 0.26)].

## Measures of message effectiveness

**Perceived message effectiveness.** This study included two dependent variables as indicators of message effectiveness. First, a perceived message effectiveness (PME) score measured the extent to which participants believed the quote would be persuasive in encouraging themselves and other parents to vaccinate their children. PME has been validated as a strong predictor of behavioral intention and behavior [44, 45]. A six-item scale summing raw scores (alpha = 0.86) measured PME [e.g., *This quote will be effective in encouraging parents to get their children vaccinated against COVID-19, This quote is unlikely to convince a parent to get their child a COVID-19 vaccine* (reverse coded)]. Response options ranged from 1 (lowest perceived effectiveness) to 6 (highest perceived effectiveness). Items were adapted from a previously validated scale [46] for the current study context. Supplementary materials detail modifications (S1B Table in S1 File) and summarize results of principal components analysis for the created scale.

**Pro-vaccine beliefs.** The second outcome measure was beliefs about getting one's child vaccinated against COVID-19. A pro-vaccine belief scale (alpha = 0.94) measured five beliefs about the safety and efficacy of getting children vaccinated against COVID-19 [e.g., *Getting the COVID-19 vaccine benefits my child, The COVID-19 vaccine is unsafe for children* (reverse coded)]. The scale ranged from 1 (most anti-vaccine) to 7 (most pro-vaccine). Items were adapted from the vaccine hesitancy scale developed by the SAGE Working Group on Vaccine Hesitancy [47] and later validated by Shapiro and colleagues [48]. All survey measures are included in supplementary material (S1 File).

## Additional measures

Two measures were included to gather relevant background information about the sample: participant influence over whether their child receives the vaccine (decision-maker) and child COVID-19 vaccination status. These items were only used as descriptive variables to describe the participating samples, not as eligibility criteria for the study or in the test of the hypotheses.

**Decision-maker.**   The decision-maker item asked participants whether they were responsible for deciding which vaccines their youngest child receives (either independently or together with another parent).

**Child vaccination status.**   To measure child vaccination status, participants were asked whether they had at least one child who was eligible for a COVID-19 vaccine (at the time of study, 5-17-years-old) and had not received the vaccine. To control for potential social desirability bias, participants were randomly selected to receive the vaccine status question pre-treatment (51.3%; $n$ = 204) or post-treatment (at the end of the survey) (48.7%; $n$ = 194).

## Analysis

All analyses were conducted in Stata version 15.0. A between-subjects 2 (trust cue vs. no trust cue) X 2 (pro-vaccine evidence vs. no pro-vaccine evidence) analysis was conducted to test the main effects of trust (H1), pro-vaccine evidence (H2), and their interaction on PME and pro-vaccine beliefs (H3). To assess simple main effects, pairwise comparisons of marginal linear predictions and calculations of effect size (Cohen's $d$) were used to measure the difference in outcomes between each treatment condition and the control condition.

A two-way ANOVA was used to test interaction effects. Prior to analyses, assumptions of ANOVA were evaluated. Others have shown a skew less than |2.0| and kurtosis less than |9.0| satisfy the assumption of normality for ANOVA [49]. The current data met these criteria (skewness: PME, -0.38 and pro-vaccine beliefs, -0.58; kurtosis: PME, 2.40 and pro-vaccine beliefs, 2.28). A Levene's F test confirmed the assumption of homogeneity was also satisfied [PME: $F(3, 394)$ = 1.68, $p$ = 0.172 and pro-vaccine beliefs: $F(3, 394)$ = 1.52, $p$ = 0.209]. The experimental design ensured independence across groups. Thus, the assumptions of normality, homogeneity, and independence were satisfied.

## Results

### Descriptive data

In total, 398 participants completed the survey [trust cue-only condition $n$ = 93; pro-vaccine evidence-only condition $n$ = 102; trust cue + pro-vaccine evidence condition $n$ = 102; no trust cue + no evidence (control) $n$ = 101]. The three participants who did not complete the survey were excluded from analysis. Demographic data for all participants is reported in Table 1. As shown, 65% of participants were the parent or guardian of at least one unvaccinated child. Nearly all participants (97%) were responsible for deciding which vaccines their child receives (decision-maker), as either the primary decision-maker (68%) or as a co-decision-maker (29%).

### Main effects of trustworthiness and pro-vaccine evidence on PME and pro-vaccine beliefs

It was hypothesized there would be main effects of trustworthiness (H1) and pro-vaccine evidence (H2) on message effectiveness (PME and pro-vaccine beliefs). Table 2 provides the mean PME and pro-vaccine belief scores for each condition. Supporting H1 and H2, pairwise comparisons showed each treatment had a significant effect compared to the control group [trust-only vs. control condition: PME: $t(394)$ = 3.83; $p < .001$; pro-vaccine belief: $t(394)$ = 2.77; $p < .001$; pro-vaccine evidence-only vs. control condition: PME: $t(394)$ = 4.07; $p < .001$; pro-vaccine belief: $t(394)$ = 2.14; $p$ = 0.033]. The effect of the combined condition (trust and pro-vaccine evidence) was also significant compared to the control ($t(394)$ = 4.36; $p < .001$).

Effect sizes were similar for all treatment conditions. Compared to the control, there were medium effects of each treatment on PME [trust: Cohen's $d$ = 0.57, 95% CI (0.29, 0.86); pro-

**Table 1. Demographic data.**

| Variable | | Count | Percentage |
|---|---|---|---|
| Gender Identity | Woman | 257 | 60.8% |
| | Man | 161 | 39.2% |
| Race/Ethnicity | White | 299 | 71.9% |
| | Black | 74 | 17.8% |
| | Asian | 18 | 4.0% |
| | Hispanic | 52 | 11.6% |
| Region | Northeast | 79 | 18.8% |
| | Midwest | 90 | 22.4% |
| | South | 174 | 42.0% |
| | West | 74 | 16.6% |
| Political party | Democratic | 154 | 36.4% |
| | Republican | 80 | 18.8% |
| | Independent | 100 | 24.1% |
| Has child age (years) | 0–1 | 40 | 10.1% |
| | 2–6 | 130 | 32.7% |
| | 7–12 | 142 | 35.7% |
| | 13–17 | 128 | 32.2% |
| Other | Respondent age (M/SD) | — | 42.6 (10.5) |
| | Decision-maker | 387 | 97.2% |
| | Has unvaccinated child | 257 | 64.6% |
| *N* | | 398 | 100% |

*Note.* All demographic variables except for decision-maker and child vaccine status were collected a priori as part of participation in the Forthright panel.

vaccine evidence: Cohen's $d = 0.53$; 95% CI (0.25, 0.81); combined trust and pro-vaccine evidence: Cohen's $d = 0.62$; 95% CI (0.34, 0.90)] and small-to-medium effects on pro-vaccine beliefs [trust: Cohen's $d = 0.41$; 95% CI (0.13, 0.70): pro-vaccine evidence: Cohen's $d = 0.29$; 95% CI (0.01, 0.56); combined trust and pro-vaccine evidence: Cohen's $d = 0.36$; 95% CI (0.09, 0.64)]. Pairwise comparisons also confirmed there were no significant differences in outcomes between any two treatment conditions.

## Interaction effects of trustworthiness and pro-vaccine evidence on PME and pro-vaccine beliefs

H3 focused on the interaction between trust cues and evidence supporting vaccines. It was hypothesized source trustworthiness and pro-vaccine evidence would each have a weaker effect

**Table 2. Mean perceived message effectiveness (pme) and pro-vaccine belief for participants in each message condition.**

| Outcome | Condition | No Trust Cue M(SD) | Trust Cue M(SD) | Total M(SD) |
|---|---|---|---|---|
| PME (1–6) | No Pro-Vaccine Evidence | 3.67 (1.28) | 4.36 (1.17) | 4.00 (1.26) |
| | Pro-Vaccine Evidence | 4.39 (1.39) | 4.44 (1.17) | 4.41 (1.28) |
| | Total | 4.03 (1.38) | 4.40 (1.15) | 4.21 (1.28) |
| Pro-Vaccine Beliefs (1–7) | No Pro-Vaccine Evidence | 4.24 (1.88) | 4.98 (1.63) | 4.59 (1.80) |
| | Pro-Vaccine Evidence | 4.80 (1.96) | 4.92 (1.85) | 4.86 (1.90) |
| | Total | 4.52 (1.94) | 4.95 (1.75) | 4.73 (1.86) |

**Table 3. Results from ANOVA testing interaction effects of trust and pro-vaccine evidence supporting vaccines on perceived message effectiveness (PME) and pro-vaccine beliefs.**

| Outcome | Source | SS | df | MS | η² | 95% CI | F | p |
|---|---|---|---|---|---|---|---|---|
| PME | Model | 39.74 | 3 | 13.24 | 0.061 | (0.012, 0.106) | 8.48 | <0.001*** |
| | Evidence | 15.50 | 1 | 15.50 | 0.025 | (0.003, 0.062) | 9.93 | 0.003** |
| | Trust Cue | 13.57 | 1 | 13.57 | 0.022 | (0.002, 0.058) | 8.69 | 0.002** |
| | Evidence X Trust | 10.11 | 1 | 10.11 | 0.016 | (0.001, 0.049) | 6.47 | 0.011* |
| | Residual | 615.32 | 394 | 1.56 | | | | |
| | Total | 655.06 | 397 | 1.65 | | | | |
| Pro-Vaccine Beliefs | Model | 33.81 | 3 | 11.27 | 0.025 | (0.000, 0.056) | 3.33 | 0.020* |
| | Evidence | 6.20 | 1 | 6.20 | 0.005 | (0.000, 0.027) | 1.83 | 0.177 |
| | Trust Cue | 18.38 | 1 | 18.38 | 0.014 | (0.000, 0.044) | 5.42 | 0.020* |
| | Evidence X Trust | 9.10 | 1 | 9.10 | 0.007 | (0.000, 0.032) | 2.69 | 0.102 |
| | Residual | 1334.92 | 394 | 3.39 | | | | |
| | Total | 1368.73 | 397 | 3.45 | | | | |

*$p < .05$;
**$p < .01$;
***$p < .001$

on message effectiveness when the other was excluded from the message. Consistent with this interaction hypothesis, as described in the prior section, there was no significant difference between the combined (pro-evidence and trust) condition and the trust-only or evidence-only conditions in their effects on outcomes. That is, while evidence supporting vaccines and building trust each strengthened PME and pro-vaccine beliefs independently, combining these treatments did not influence these outcomes any more than either treatment on its own.

Table 3 shows the results from ANOVAs testing the interaction effects of trust cues and pro-vaccine evidence on outcomes. Supporting H3, there was a significant negative interaction between trust cues and pro-vaccine evidence in their joint effect on PME [$F(3, 394) = 6.47$; $\eta^2 = 0.02$, $p = 0.002$; 95% CI (0.01, 0.11)]. The effect of the trust cue (vs. no trust cue) on PME was weaker when pro-vaccine evidence was provided, compared to when evidence was not provided (see Fig 2a). The interaction effects of pro-vaccine evidence and trust on pro-vaccine beliefs was not significant [$F(3, 394) = 2.69$; $\eta^2 = 0.01$; $p = 0.102$; 95% CI (0.00, 0.03)], though results were directionally parallel (see Fig 2b). Thus, H3 was partially supported.

## Post-hoc sensitivity analysis

Following hypothesis tests, power levels were computed for each statistical analysis using the sample size, mean outcome scores (PME and pro-vaccine beliefs), and within-cell variances in the current study sample. The power to detect significant effects of treatments on PME exceeded 80% for main effects (trust: ß = 0.84; evidence: ß = 0.89) and was slightly below 80% for the interaction of trust and evidence (ß = 0.72). The power to detect significant effects of each treatment on pro-vaccine beliefs was below 80% for both main and interaction effects (trust: ß = 0.64; evidence: ß = 0.27; interaction of trust and evidence: ß = 0.39). This post-hoc analysis suggests the study's sample size was appropriate for detecting treatment effects on PME but may have been too low to detect some significant main and interaction effects on pro-vaccine beliefs.

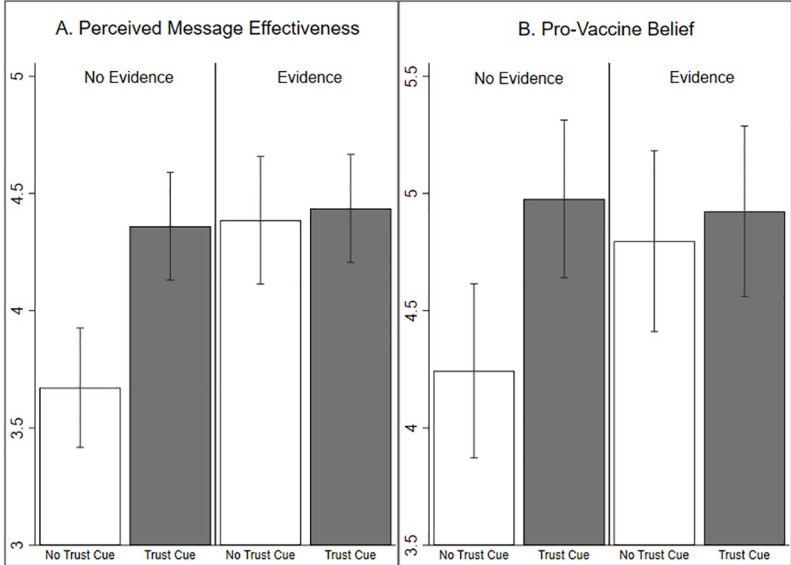

**Fig 2. Message effectiveness by trust cue versus no trust cue and inclusion of scientific evidence supporting vaccines versus no evidence.** Bar graph illustrating the mean perceived message effectiveness (PME) score (2a) and mean pro-vaccine belief score (2b) for participants in each condition, with 95% confidence intervals. PME was measured using a six-item scale ranging from 1–6. Pro-vaccine belief was measured using a five-item scale ranging from 1–7.

## Discussion

To the author's knowledge, this is the first experimental study to test the effects of source trustworthiness, evidence supporting vaccines, and their interaction with COVID-19 child vaccine recommendation messages for parents. Supporting main effects hypotheses, both building trust in the recommendation source and including pro-vaccine evidence significantly strengthened message effectiveness compared to a control message. Further, supporting the dual-processing hypothesis, the effectiveness of each of these strategies was higher independently than in combination with the other. That is, the persuasive effect of a trust cue was stronger when pro-vaccine evidence was not included in the message, and the persuasive effect of pro-vaccine evidence was stronger when the trust cue was not included. The direction of these interaction effects was parallel for both outcomes [perceived message effectiveness (PME) and pro-vaccine beliefs]; it was statistically significant for PME.

Results suggest trust in recommendation sources may influence parent vaccine decisions most when the information to make evidence-based judgments is absent and that including evidence supporting vaccines may reduce the effects of source trust. These findings align with prior studies that have found source credibility may have conditional effects on persuasive outcomes and expands this literature to the context of parent COVID-19 vaccine decisions [29, 30, 33, 36]. Most conceptually and topically similar to the present study, a longitudinal observational study of U.S adults found the association between trust in expert recommendation sources and subsequent COVID-19 vaccination was attenuated by belief in the efficacy and safety of vaccines [36]. The results from the present experimental study of parents are parallel—demonstrating the association between perceived trustworthiness of an expert source and pro-vaccine outcomes was attenuated by evidence in favor of the efficacy and safety of vaccines. The use of experimental methods builds on prior research and strengthens causal claims.

Findings support dual-processing theories, suggesting that individuals are most influenced by source trustworthiness (a heuristic cue) in the absence of highlighting evidence supporting a recommendation (i.e., information to support systematic processing) [12]. Critics of dual-processing theories have argued that the model is only useful for explaining attitude formation in hypothetical scenarios, rather than attitude formation in the context of real health recommendations [27, 50]. The present study challenges this critique. Given the plethora of information and misinformation in the media about COVID-19 vaccines [51, 52], it is likely participants entered the study aware of the conversation around this issue. The present study's finding, that pro-vaccine evidence and source trustworthiness each had a stronger effect on parent vaccine endorsement when the other was absent, suggests dual-processing may indeed play a role in influencing perceptions about real-world guidelines.

## Strengths and limitations

This study has several strengths, including the use of experimental methods to strengthen internal validity, examination of dual-processing theory in a new context, and relevance to current COVID-19 communication priorities. However, a limitation of this study is the use of a single message per condition. While manipulation checks showed independent variables were successfully manipulated by the single-message treatment, including additional message exemplars would help distinguish between effects of specific messages and effect of the constructs they represent [53, 54]. Further, the combined trust + pro-vaccine evidence condition included trimmed-down versions of each cue and the control condition respondents received only a brief message. This may have limited the ability to compare conditions. Future work could address this limitation by exposing participants in each condition to multiple messages with varying lengths.

In addition, the negative interaction effect of trust and pro-vaccine evidence was significant on PME, but not on pro-vaccine beliefs. It is possible the null hypothesis is true, and that trust and pro-vaccine evidence do not have interaction effects on pro-vaccine beliefs. However, given the direction of effects was parallel to those on PME and close to significant, null effects may also be explained by one of a few methodical limitations. First, the sample size was limited by budget constraints and may have been too small to detect significant effects. Post-hoc power estimates suggest a sample size closer to 1,100 would have been sufficient to detect an effect on pro-vaccine beliefs at 0.05 significance level with 80% power. Second, compared to PME, pro-vaccine beliefs may be too difficult an outcome to alter in the short-term context of a lab experiment. PME has been validated as an antecedent of subsequent belief change and may be a stronger short-term indicator of message effects [45]. Measuring pro-vaccine beliefs following multiple message exposures over time might more accurately captured interaction effects on belief-change. Third, attention checks were not included in this study. It is possible that limiting analyses to participants who passed attention checks would have increased likelihood of detecting effects on pro-vaccine beliefs. Finally, it is worth noting that all conditions, including the control, conveyed information supporting child vaccination. Therefore, responses may have been influenced by social desirability bias. Future work should control for social desirability.

Despite these limitations, the detection of significant main effects on pro-vaccine beliefs and main and significant interaction effects on PME should be considered strong indicators of the potential influence of both trust and exposure on parent vaccine decisions. Prior literature has shown behavioral beliefs strongly predict subsequent behavior for a range of contexts [55], including COVID-19 vaccination [56, 57]. PME has also been validated as a strong predictor of behavioral beliefs, intention, and behavior [44, 45]. Therefore, though this study did not

measure behavior change, treatment effects on behavioral beliefs and PME—both antecedents of behavior—are promising indicators of influence on subsequent child vaccination decisions.

## Implications and future directions

Some frameworks have been developed for guiding experts in their discussions of child vaccination with parents [58–60]. For example, the C.A.S.E (Corroborate, About Me, Science, and Explain/advise) framework suggests first building trust by acknowledging specific parent concerns and then describing scientific evidence in support of the recommended vaccine [60]. This strategy assumes focusing on both establishing trust and providing information about vaccination outcomes will be most effective in reducing parent vaccine hesitancy. However, this framework was established to guide interpersonal communication and has not been empirically tested in the context of developing media messages [14]. It is possible that a more useful approach for message design is to focus on *either* building trust *or* explaining pro-vaccine evidence.

The findings of this study—that one type of perception may attenuate the influence of another type of perception on recommendation acceptance—may have relevance for health recommendations beyond child COVID-19 vaccination. As COVID-19 guidelines evolve, with the introduction of updated booster formulas and treatments, the potential role of dual-processing may be useful to consider when developing health messages. On the one hand, highlighting evidence supporting recommended behaviors and strengthening confidence in behavioral benefits may be particularly effective among those whose trust in experts is low. For example, public health messages might argue that vaccine booster doses reduce the risk of serious infection or transmission of new variants. These evidence-based arguments may appeal to individuals who are motivated to engage in deliberative processing, even if they distrust public health sources [61].

On the other hand, there are situations (such as during the first years of the COVID-19 pandemic) in which evidence is new or complex and experts may find it challenging to increase the public's confidence in the positive outcomes of recommended behaviors. To a public that encounters a plethora of information each day from a complex media environment, scientific knowledge about the pandemic and recommended behaviors may feel inaccessible or difficult to process. When individuals are uncertain about the efficacy and safety of recommended behaviors, sources with whom trust has been established can help encourage behavior even in the absence of easily presented evidence [62]. For this reason, efforts to rebuild and maintain trust in experts should remain a public health priority.

In the current study, an expert source self-identified as a parent and described vaccination as a means to optimize their own children's health and well-being. This message significantly strengthened parent trust in the source. The measure of trust was not topic-specific; items pertained to the feeling that the source was honest, sincere, and shared the participants' concerns for their children. Therefore, the type of trust appeal applied here may increase acceptance of a broad array of child health recommendations. A similar values-based approach may be effective for experts seeking to build parent trust when providing other health recommendations for children.

## Conclusions

The findings of this study have important implications for health communication. First, results show it is possible for public health messages to influence recommendation acceptance by either emphasizing the scientific evidence supporting the recommendation or building trust in the source. Second, combining messaging approaches might not be necessary to optimize

persuasive outcomes. Pro-recommendation evidence and source trustworthiness may each be most effective when the other is absent. As COVID-19 guidelines evolve, individuals may feel unsure about the safety and efficacy of engaging in recommended behaviors and look to trusted sources for guidance. Establishing trust in expert sources may help encourage recommendation-following in this context of uncertainty. On the other hand, when communicating with those whose trust in experts is low and difficult to alter, clearly describing scientific knowledge supporting recommended behaviors may be an effective strategy for encouraging recommendation acceptance.

## Supporting information

**S1 File. Treatments and measures.**
(DOCX)

**S2 File. Raw data and codebook.**
(ZIP)

## Acknowledgments

Thank you to Robert Hornik and Diana Mutz, who provided expertise and support throughout all aspects of this study. I also thank Emma Jesch, Danielle Clark, and Chioma Woko for their feedback on this project.

## Author Contributions

**Conceptualization:** Ava Irysa Kikut.

**Data curation:** Ava Irysa Kikut.

**Formal analysis:** Ava Irysa Kikut.

**Funding acquisition:** Ava Irysa Kikut.

**Investigation:** Ava Irysa Kikut.

**Methodology:** Ava Irysa Kikut.

**Project administration:** Ava Irysa Kikut.

**Visualization:** Ava Irysa Kikut.

**Writing – original draft:** Ava Irysa Kikut.

**Writing – review & editing:** Ava Irysa Kikut.

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
