## [Decision Letter · Decision Letter 0]

12 Aug 2022

PONE-D-22-10305The doctor knows or the evidence shows: An online survey experiment testing the effects of source trust, pro-vaccine evidence, and dual-processing in expert messages recommending the child COVID-19 vaccination to parentsPLOS ONE

Dear Dr. Kikut,

Thank you for submitting your manuscript to PLOS ONE. After careful consideration, we feel that it has merit but does not fully meet PLOS ONE’s publication criteria as it currently stands. Therefore, we invite you to submit a revised version of the manuscript that addresses the points raised during the review process.

The manuscript has been evaluated by two reviewers, and their comments are available below. The reviewers have raised a number of major concerns. They state that there is a general lack of detail regarding crucial aspects of the sample and the methodology, and raise concerns that the statistical analyses described in the manuscript are inadequate.   Could you please carefully revise the manuscript to address all comments raised?

We look forward to receiving your revised manuscript.

Kind regards,

Maria Elisabeth Johanna Zalm, Ph.D

Editorial Office

PLOS ONE

Journal Requirements:

Reviewers' comments:

Reviewer's Responses to Questions

**Comments to the Author**

1. Is the manuscript technically sound, and do the data support the conclusions?

Reviewer #1: Partly

Reviewer #2: Partly

2. Has the statistical analysis been performed appropriately and rigorously? 

Reviewer #1: No

Reviewer #2: No

3. Have the authors made all data underlying the findings in their manuscript fully available?

Reviewer #1: Yes

Reviewer #2: Yes

4. Is the manuscript presented in an intelligible fashion and written in standard English?

Reviewer #1: No

Reviewer #2: No

5. Review Comments to the Author

Reviewer #1: This manuscript reports on the results of an online study which employs an experimental design to test whether the presentation of COVID-19 vaccine information to parents, including source trustworthiness and evidence of vaccine efficacy, differentially and interactively impact self-reported intentions to vaccinate their child and belief regarding vaccine safety and efficacy for children. The topic is highly relevant, and the study design is novel. Additionally, the author is commended for pre-registering their hypotheses prior to data collection. However, there are significant concerns throughout the manuscript that need to be addressed. Briefly, these include a lack of appropriate scientific language, inadequate statistical analyses, and a general lack of detail regarding crucial aspects of the sample and methodology. Please see attached comments.

Reviewer #2: The author addressed the problem of COVID-19 vaccination decision for children in the theoretical framework of dual-processing theory. The paper quotes mostly previous empirical researches related to the effects of 'evidence' and 'trustworthiness' with some theoretical literature related to dual processing theory, however it lacks the broader theoretical background of vaccine hesitancy. The empirical analysis based on experiment using online survey as a tool, said to be national probability sample. However the given source can not be assessed according to the applied sampling methodology (also the randomized experiment doesn't necessarily require probability sample). The author used descriptive statistics and ANOVA for analyzing data, however the paper doesn't include any tests for assumption of ANOVA, and it doesn't contain measures of effects. The author drove conclusions based on the statistical data and the previously quoted literature, however it also should address the generalizability of the current research

The paper well structured. It contains two tables with descriptive statistics, which are also quoted in the text, while doesn't include result of the ANOVA analysis in table. The paper quotes 48 sources in the order of appearance.

The strength of the paper is that it address an important problem, it is based on well-designed research and it has an important conclusion. However, the paper needs further improvement to be more evident.

Major issues to be addressed by the author are the following:

- further literature about vaccine hesitancy should be included, that embeds the current research in a broader context

- sampling method should be clarified

- tests for assumptions of ANOVA should be conducted and evaluated

- a measure of effect size should also be reported in the case of ANOVA

- the conclusion should include some thoughts about how the current results can be generalized

Further minor issues:

- I would advise combining descriptive statistics into one table, and including one further table with the results of the ANOVA

- Title for tables should be revised to be more informative and according to APA rules

- Wording of the paper should be checked, it would be better to avoid sentences in the first person

6. PLOS authors have the option to publish the peer review history of their article (what does this mean?). If published, this will include your full peer review and any attached files.

Reviewer #1: No

Reviewer #2: No

---

## [Author Response · Author response to Decision Letter 0]

26 Sep 2022

Dear Reviewers, 

Thank you for your thoughtful feedback and the opportunity to revise this manuscript entitled “The doctor knows or the evidence shows: An online survey experiment testing the effects of source trust, pro-vaccine evidence, and dual-processing in expert messages recommending the child COVID-19 vaccination to parents”. Below I provide a point-by-point response to your suggestions. In addition to the clean updated manuscript, I also provide a version with tracked changes. Your suggestions have helped to improve this manuscript, and I hope we have addressed your concerns. 

Thank you again for your careful review. 

Comments from the Reviewers: 

Reviewer #1

Reviewer: This manuscript reports on the results of an online study which employs an experimental design to test whether the presentation of COVID-19 vaccine information to parents, including source trustworthiness and evidence of vaccine efficacy, differentially and interactively impact self-reported intentions to vaccinate their child and belief regarding vaccine safety and efficacy for children. The topic is highly relevant, and the study design is novel. Additionally, the author is commended for pre-registering their hypotheses prior to data collection. However, there are significant concerns throughout the manuscript that need to be addressed. Briefly, these include a lack of appropriate scientific language, inadequate statistical analyses, and a general lack of detail regarding crucial aspects of the sample and methodology. Please see attached comments.

Response: Thank you for your feedback. I hope I have addressed these concerns. 

OVERALL/WRITING 

1. Reviewer: Several grammatical errors were identified throughout the manuscript (e.g., lines 34-35). Please revise carefully to ensure it is free from such errors. 

Response: Thank you. The manuscript has been carefully revised to remove grammatical errors. 

2. Reviewer: I would encourage the author to review style guides for scientific writing. The manuscript would be greatly improved if for example instances of first person “I” were removed, and rhetorical questions were changed to statements (especially in the introduction). 

Response: Thank you for this feedback. All first-person statements have been revised to third-person. All rhetorical questions have been changed to statements. 

3. Reviewer: There is a lack of consistent language when referring to specific constructs throughout the manuscript. For example, the author refers to the interaction between peripheral and central cues on line 108, but earlier describes them as type 1 and type 2 processing. 4. When referring to provide evidence on vaccine efficacy/safety, I would encourage the author to explicitly state this at very mention even if it’s “vaccine evidence” just to make it clear for the reader. 

Response: Thank you. Language has been edited for consistency. 

ABSTRACT

1. Reviewer: Please consider revising the hypothesis and design section of the abstract to improve clarify. For example, presenting to the reader that you are interested in the effects of x on x descriptively, rather than stating that you are interested in the main effects and interactions (line 17). Additionally, referring to aspects of the design in an abbreviated manner (e.g., “trust cue vs. no trust cue” on line 19) can be confusing as this has not yet been explained to the reader. 

Response: Thank you for this point. The abstract has been revised significantly to improve clarity.

2. Reviewer: Please use the term significant when describing a main effect or interaction (if appropriate) as simply stating that there was a main effect or interaction does not provide adequate detail to the reader. 

Response: Thank you. Language has been revised in the abstract and the manuscript body to specifically note significance (or non-significance). 

INTRODUCTION 

1. Reviewer: Some of the writing in the introduction is unclear or lacks detail. For example, on line 42, the author states “how do the effects of source trust differ” but does not name the outcome variable only the moderator (present of evidence). On line 94, the author does not specify the directionality of the relationship named. Please revise the entire manuscript to ensure that statements regarding the relationships among variables or constructs are complete and clear to the reader. 

Response: Thank you for this note. The introduction has been updated to improve clarity. In addition, the interaction hypothesis (H3, line 149) has been revised to reflect that trust may reduce the need for evidence and evidence may reduce the need to build trust. I have adjusted the wording throughout the abstract, introduction, and discussion to show moderation could work in either direction. The data does not show which variable moderates the effect of the other. 

2. Reviewer: On lines 66-76, the author describes the construct of source credibility. Overall, this section is a bit confusing to the reader (stating that credibility can be broken into trustworthiness and perceived expertise and that trustworthiness can be further broke into trustworthiness and goodwill). Finally, there is the addition of including measures of honestly and empathy at the end. I would encourage the author to make this section much more concise and describe only what is necessary to describe what is measured. Additionally, more rationale should be given for the inclusion and purpose of honestly and empathy. 

Response: Thank you. This section has been shortened to only include information that is relevant to the reader and the purpose of manipulating intention-based trust (honesty/empathy) is now included (lines 88-99): 

Prior research has found vaccine hesitant parents are more likely to believe health care providers are pressured by vaccine manufacturers to recommend vaccines and are motivated by self-interest (16). Experts may reduce these biases by demonstrating their motivation to provide recommendations aligning with children’s best interests. Recognition of shared values has been shown to build trust in a source’s intentions (i.e., source trustworthiness) (21). Trust in a source can increase attention to messages (22) and strengthen message effects on beliefs (23,24). Specific to COVID-19 vaccine messaging, there is some evidence source trustworthiness increases message effectiveness (25). In the context of child vaccines, trusting the recommendation source’s motivations has been shown to decrease parent hesitancy (26).

METHODS 

1. Reviewer: The Forthright Capabilities panel should be explained in detail. Is this a labor market platform similar to MTurk? How are participants recruited? How many are in the pool? How was this study advertised to them? Also, the author mentioned that the demographic data matches the US population but does not provide any of this data. Please include descriptive statistics for age, gender, sex, ethnicity, race, education, etc. as available. These are crucial aspects to consider given the nature of this study and potential application to different groups within the US. 

Response: Yes, Forthright Capabilities is a labor market platform similar to MTurk. I have included additional details about the panel and recruitment process in the Sample section, lines 155-176 (see quote in bold below). I have also added a new table (Table 1) to provide descriptive statistics on the sample that were made available through the Forthright platform.

An online survey experiment was programmed in QualtricsXM and administered through the Forthright Capabilities panel (39). Forthright is an online research panel that recruits panellists through digital advertising and mail campaigns using addressed-based probability sampling. Panellists are screened with security checks and asked to provide basic demographic information for a profiling database upon enrolment in the panel. The researcher provides Forthright with information on eligibility criteria and sample size. When the survey is live, Forthright alerts potentially eligible respondents via email, offering information about the survey length and compensation. Once the sample size target is met, or slightly exceeded, the survey is closed to panelists. 

Participants were eligible for the current study if they resided in the U.S. and were parents or guardians of at least one child under 18 years of age. In total, 420 Forthright panelists received an invitation to participate in the study (out of a pool of approximately 5000 eligible panelists) through stratified random sampling using nationally representative quotas for gender, age, region, and ethnicity. Of those invited, 401 panelists participated (response rate=95.5%). Due to budget constraints, the survey experiment was conducted in two phases. An initial sample of 200 participants were recruited to complete the experiment between 12/9-12/11/2021 (n=200). An additional 201 participants were recruited to complete the same experiment between 1/25-1/29/2022 (n=201). The samples were combined.

2. Reviewer: The author mentioned on line 136 that to increase statistical power, the two samples were combined. Was a power analysis conducted? If not, how does the author justify the sample size? 

Response: Though power calculations would have called for a larger sample size, the sample size was determined by budget constraints. I was a student in an academic course that offered funding to recruit up to 200 participants for a class assignment. After completing the course, I acquired funding to recruit an additional 200 participants. This additional data did not change results. Any result that is significant (or not) in the larger sample was also significant (or not) in the initial sample. I added an extra note to briefly explain this (line 169): Due to budget constraints, the survey experiment was conducted in two phases.

3. Reviewer: On line 135, the author mentions that the survey was conducted through an academic course but does not provide further detail. Please explain how participants were recruited and how they qualified within the course. Also, was this a course at a given university or other institution? 

Response: Participants were recruited through Forthright, as described above. Funding to recruit through the Forthright platform was provided by the professor of an academic course to support final projects. The way the section was previously written unintentionally suggested the participants were students in the course. I have updated the section to address this.

4. Reviewer: In the Design section, the author names the conditions (e.g., “trust cue vs. no trust cue”) prior to explaining what they are. To provide clarity to the reader, please describe the conditions first, then the analytic design second. 

Response: Thank you for this suggestion. I have moved the sentence describing the analytic design below, to the Analysis section (see line 264): A between-subjects 2 (trust cue vs. no trust cue) X 2 (pro-vaccine evidence vs. no pro-vaccine evidence) analysis was conducted…

5. Reviewer: Lines 140 to 162 lack adequate detail and clarity regarding the procedure. Please revise to guide the reader step-by-step through how participants started the experiment survey, including where they were, what survey software was used, etc. A figure would help clarify the design of this study to clearly identify each step of the procedure and where the conditions branched off. Regarding lack of detail, the author should provide samples of the materials presented (e.g., naming the specific measures on line 146, including short quotes of each cue type, etc.). 

Response: I have added additional detail about how participants started the survey and the step-by-sept procedure (lines 178-187):

After opening the survey on their Forthright dashboard and agreeing to participate, respondents were presented with one of four quotes recommending COVID-19 vaccination. All quotes were attributed to the same fictional physician (Dr. Taylor Clark). To help ensure the whole quote was read, respondents were not shown the button allowing for progression to the next page for the first 30 seconds the quote was displayed. After reading the quote, all participants were asked the same set of questions, including outcome measures and manipulation check measures. The mean response time for the experiment was 3.3 minutes.

I have included a sentence about the survey software used on line 155 (“An online survey experiment was programmed in QualtricsXM and administered through the Forthright Capabilities panel”). I have also added the names of outcome measures (“PME and vaccine beliefs”) and manipulation check measures (“perceived trustworthiness of quote source and whether quote included evidence”) on line 184-186 Additionally, I included a new figure (Figure 1) describing the survey procedure. 

6. Reviewer: The author states that the trust + evidence condition utilized a trimmed down cue. How was it trimmed down? The author should mention why this decision was made (beyond keeping the time the same) in the discussion, especially given that changing this cue could potentially introduce error or reduce the ability to truly compare conditions. 

Response: Thank you for this point. The following text has been added to the “Strengths and limitations” section of the discussion (lines 394-398):

The combined trust + pro-vaccine evidence condition included trimmed-down versions of each cue and the control condition respondents received only a brief message. This may have limited the ability to compare conditions. Future work could address this limitation by exposing participants in each condition to multiple messages with varying lengths.

7. Reviewer: On line 164, the author mentions that the manipulation check “ensured” – please revise as manipulation checks are only limited post-hoc examinations that estimate whether the manipulation may have had the intended effect. 

Response: Thank you for this note. I updated this sentence to say “A manipulation check assessed whether trust cues…” rather than ensured (line 209). 

8. Reviewer: The method of describing different measures in the methods section is a bit confusing to the reader. I would recommend breaking it down with subheadings for each measure (in the order they were administered) and then describing the order of administration in the earlier design section. Please provide psychometric information as available for each measure (e.g., internal consistency, test re-test reliability, etc.) and whether it has been validated previously. Additionally, please provide an example item for each measure in quotes.

Response: Thank you for this feedback. I have included example items for each measure described. I have also updated the Measures of message effectiveness section to include the sub-headers, Perceived message effectiveness and Pro-vaccine beliefs. The order of measures in the survey is now shown in Figure 1. Manipulation checks were administered last to avoid spillover effects on outcome measures. However, I describe the results of manipulation checks directly after describing the treatment for conceptual consistency—the manipulation checks help show independent variables were successfully manipulated by the treatment. The internal consistency is included for all measures perceived trustworthiness (alpha=0.89, line 211), perceived evidence strength (shows correlation rather than 

 alpha because only two-items included: r=0.52, line 222), PME (alpha=0.86, line 235), pro-vaccine belief (alpha = 0.94, line 243). Citations are included to show scales have been previously validated. Because this is a single-wave study, I was unable to measure test-retest reliability. 

9. Reviewer: The author mentions that the two measures regarding child vacation status and control over vaccine choice were split (50% at the beginning, 50% at the end). How could this have differentially impacted groups? The author should re-run the analyses controlling for this as a variable to rule out the introduction of error. 

Response: By randomly assigning respondents (across all four treatment groups) to receive the child vaccination status prior to or following the study, I controlled for any effect the question may have had. Respondents in all treatment groups had an equal likelihood of having responded to the question so any effect would be cancelled out. To illustrate this, I have run the interaction ANOVA controlling for child vaccination question order (shown below). As shown, results were not affected when this variable was included in the model. 

Outcome Source SS df MS η2 95% CI F p

Perceived Message Effectiveness Model 40.36 4 10.09 0.062 (0.018, 0.104) 6.45 0.00

 Evidence 15.54 1 15.54 0.025 (0.003, 0.062) 9.94 0.00

 Trust Cue 13.45 1 13.45 0.021 (0.002, 0.057) 8.60 0.00

 Evidence X Trust 9.83 1 9.83 0.016 (0.001, 0.048) 6.29 0.01

 Child vaccination question asked pre (vs. post) survey 0.621 1 0.621 0.001 (0.000, 0.017) 0.40 0.53

 Residual 614.70 393 1.56 

 Total 655.06 397 1.65 

Pro-vaccine Beliefs Model 34.10 4 8.53 0.025 (0.000, 0.054) 2.51 0.04

 Evidence 6.22 1 6.22 0.005 (0.000, 0.027) 1.83 0.18

 Trust Cue 18.27 1 18.27 0.014 (0.000, 0.044) 5.38 0.02

 Evidence X Trust 8.91 1 8.91 0.007 (0.000, 0.031) 2.69 0.10

 Child vaccination question asked pre (vs. post) 0.297 1 0.297 0.000 (0.000, 0.011) 0.09 0.77

 Residual 1334.92 393 3.40 

 Total 1368.73 397 3.45 

10. Reviewer: The analytic plan section should be expanded with more detail (e.g., what software was used?). The author should also describe the plan for making pairwise comparison following each main effect or interaction. 

Response: Thank you for this feedback. I have updated the Analysis section to include the statistical software (Stata, line 264). I add additional detail to the analysis section, including description of planned contrasts (line 267-270) and tests of assumptions of ANOVA (line 271-278). 

RESULTS 

1. Reviewer: On line 213, the author mentions “65% of participants” but is this out of the total 398? If so, would this bring the final sample to 258? Please clarify. 

Response: Child vaccination status was included as a descriptive measure rather than as an eligibility criterion. I wanted to ensure there were at least some portion of participants whose children were eligible for the vaccine. Given behavior was not an outcome measure, I did not need to exclude parents whose children were already vaccinated. To clarify, I added the following sentence under Additional Measures (line 260): These items were only used as descriptive variables to describe the participating samples, not as eligibility criteria for the study or in the test of the hypotheses.

2. Reviewer: For tables 1-2, there is no need to indicate what M and SE stand for as it can be assumed the reader knows. Additionally, there is no need to state that the measure contained six items from 1-6. 

Response: Thank you. These notes have been deleted from the table captions. 

3. Reviewer: On line 223, the author states that all manipulation conditions had higher means than the control, but was this a significant difference? If not, this may mislead the reader. Please clarify. 

Response: The means for the manipulation conditions were significantly higher than the that of the control based on pairwise comparisons. Taking into account your suggestion to include results of pairwise comparisons (as well as effect sizes), I have updated the analysis and results section to reflect the significant effects of each condition (vs. control). The interaction ANOVA weakened the ability to test for main effects, as it captures the effects of all conditions which include each approach (e.g., evidence with trust and evidence without trust). Therefore, the combined condition may contaminate the effects of the condition that only includes the approach of interest. I have revised the main effects sections to reflect tests of each appeal independently (trust-only vs. control and evidence-only vs control). These effects are significant for both independent approaches across outcomes. I updated the results section (300-321) to focus on this metric of main effects. I have also added the following sentence to the Analysis section (line 267-270): 

To assess simple main effects, pairwise comparisons of marginal linear predictions and calculations of effect size (Cohen’s d) were used to measure the difference in outcomes between each treatment condition and the control condition.

4. Reviewer: When describing main effects and interactions (e.g., line 234), please use the term significant to clarify. 

Response: The term significant has been added for clarity throughout this section. 

5. Reviewer: When reporting results from ANOVAs throughout the results section, please provide the F value, degrees of freedom, confidence interval, and effect size. 

Response: An additional table (Table 3) has been added to show ANOVA results with this information. In addition, all of the above statistics have been added to within-text reports of results. 

6. Reviewer: It does not appear that the author conducted pairwise comparisons (if so, this is unclear) but merely provided a descriptive comparison of means. As it stands, comparing means without any type of test is not a statistically valid way to make conclusions from data. Please clarify. 

Response: I have updated the analysis section to clarify that pairwise comparisons were conducted (see response to comment #3 in this section). 

7. Reviewer: As mentioned previously, please include effect sizes for every statistical test run. 

Response: I have added effects sizes (Cohen’s d) for all pairwise comparisons as well as η2 in the ANOVA table (Table 3). I have also included standard deviations (in the place of previously reported standard errors) so the reader can calculate standardized effect sizes. 

8. Reviewer: At several points, the author mentions that there are results that do not reach “traditional significance.” This is a massive concern as it implies to the reader that the effect was there but simply didn’t meet a certain threshold. This is not the case as any effect that does not reach significance must be attributed as no different than random. Please remove all instances of this phrase throughout the results and discussion and use explicit language to identify that in these cases, there was no effect found. 

Response: Thank you for this point. The wording used to describe non-significant results has been updated throughout the manuscript. In addition, I have added a paragraph to the “Strengths and limitations” section of the discussion to reflect on the insignificant interaction result (line 399-412): 

In addition, the negative interaction effect of trust and pro-vaccine evidence was significant on PME, but not on pro-vaccine beliefs. It is possible the null hypothesis is true, and that trust and pro-vaccine evidence do not have interactive effects on beliefs. However, given the direction of effects was parallel to those on PME and close to significant, null effects may also be explained by one of two methodical limitations. First, the sample size was limited by budget constraints and may have been too small to detect significant effects. Post-hoc power estimates using the mean pro-vaccine belief scores and error variance from the current sample suggest a sample size closer to 1,100 would have been sufficient to detect an effect at 0.05 significance level with 80% power. Second, compared to PME, beliefs may be too difficult an outcome to alter in the short-term context of a lab experiment. PME has been validated as an antecedent of subsequent belief change and may be a stronger short-term indicator of message effects (45). A follow-up study following multiple message exposures might have more accurately captured interaction effects on belief-change.

DISCUSSION 

1. Reviewer: Please revise the limitations section to include all of the aforementioned concerns in this review that cannot be addressed.

Response: The “Strengths and limitations” section of the discussion has been expanded to include a discussion of the trimmed-down versions of each cue and the non-significant interaction result (lines 386-412). 

Reviewer #2

Reviewer: The author addressed the problem of COVID-19 vaccination decision for children in the theoretical framework of dual-processing theory. The paper quotes mostly previous empirical researches related to the effects of 'evidence' and 'trustworthiness' with some theoretical literature related to dual processing theory, however it lacks the broader theoretical background of vaccine hesitancy. The empirical analysis based on experiment using online survey as a tool, said to be national probability sample. However the given source can not be assessed according to the applied sampling methodology (also the randomized experiment doesn't necessarily require probability sample). The author used descriptive statistics and ANOVA for analyzing data, however the paper doesn't include any tests for assumption of ANOVA, and it doesn't contain measures of effects. The author drove conclusions based on the statistical data and the previously quoted literature, however it also should address the generalizability of the current research

The paper well structured. It contains two tables with descriptive statistics, which are also quoted in the text, while doesn't include result of the ANOVA analysis in table. The paper quotes 48 sources in the order of appearance. The strength of the paper is that it address an important problem, it is based on well-designed research and it has an important conclusion. However, the paper needs further improvement to be more evident.

Major issues to be addressed by the author are the following:

1. Reviewer: Further literature about vaccine hesitancy should be included, that embeds the current research in a broader context

a. Response: Thank you for this feedback. Two paragraphs have been added at the beginning of the literature review to provide broader context on vaccine hesitancy (lines 60-76). 

Outside of COVID-19 vaccination, several studies have examined vaccine hesitancy among parents. One of the most common types of concerns among hesitant parents is vaccine safety (11). For example, parents who are vaccine hesitant may believe vaccines will weaken their children’s immune system (12–14) or lead to serious side effects (12,13,15). Another class of concerns center around vaccine efficacy and necessity (11); vaccine hesitant parents may be more likely to question the effectiveness of vaccines and to prefer their children develop immunity through natural disease rather than vaccination (12,13,16). 

Specific barriers to child COVID-19 vaccination include parent concerns about side effects, belief that the vaccine was developed too quickly to generate strong safety evidence, and the view that children are not at risk of getting seriously ill from COVID-19 (17). Factors that have been associated with parent COVID-19 vaccine acceptance include knowledge about vaccines, belief in their safety and effectiveness, high perceived risk of COVID-19, and reliance on health care providers as sources of COVID-19 information (18). Overall, the proportion of parents who show COVID-19 vaccine hesitance is higher than the proportion showing hesitance for routine child vaccines—calling for research focused specifically on this issue (18).

2. Reviewer: Sampling method should be clarified

Response: Thank you for this feedback. I have updated the Sample section (Lines 155-176) to clarify the sampling method. 

An online survey experiment was programmed in QualtricsXM and administered through the Forthright Capabilities panel (39). Forthright is an online research panel that recruits panelists through digital advertising and mail campaigns using address-based probability sampling. Panelists are screened with security checks and asked to provide basic demographic information for a profiling database upon enrolment in the panel. The researcher provides Forthright with information on eligibility criteria and sample size. When the survey is live, Forthright alerts potentially eligible respondents via email, offering information about the survey length and compensation. Once the sample size target is met, or slightly exceeded, the survey is closed to panelists. 

Participants were eligible for the current study if they resided in the U.S. and were parents or guardians of at least one child under 18 years of age. In total, 420 Forthright panelists received an invitation to participate in the study (out of a pool of approximately 5000 eligible panelists) through stratified random sampling using nationally representative quotas for gender, age, region, and ethnicity. Of those invited, 401 panelists participated (response rate=95.5%). Due to budget constraints, the survey experiment was conducted in two phases. An initial sample of 200 participants were recruited to complete the experiment between 12/9-12/11/2021 (n=200). An additional 201 participants were recruited to complete the same experiment between 1/25-1/29/2022 (n=201). The samples were combined. 

3. Reviewer: Tests for assumptions of ANOVA should be conducted and evaluated

Response: Assumptions for ANOVA were added to the Analysis section (see line 271-278). 

Prior to analyses, assumptions of ANOVA were evaluated. Others have shown a skew less than |2.0| and kurtosis less than |9.0| satisfy the assumption of normality for ANOVA (44). The current data met these criteria (skewness: PME, -0.377 and beliefs, -0.576; kurtosis: PME, 2.402 and beliefs, 2.277). A Levene’s F test confirmed the assumption of homogeneity was also satisfied [PME: F (3, 394) = 1.676, p=0.172 and Belief: F (3, 394) = 1.519, p= 0.209]. The experimental design ensured independence across groups. Thus, the assumptions of normality, homogeneity, and independence were satisfied.

4. Reviewer: Measure of effect size should also be reported in the case of ANOVA

Response: Eta squared estimates for each term have been included in the new ANOVA results table (Table 3) and in the within-text reporting of ANOVA results (lines 25, 28, 217, 227, 332, and 336). 

5. Reviewer: The conclusion should include some thoughts about how the current results can be generalized

a. Response: Thank you for this point. An additional section has been added to the discussion “Implications and future directions” which discusses how results may be generalized (lines 414-454). In addition, the “Conclusion” (line 455) section has been updated to describe broader implications for health communication beyond the study topic. 

Further minor issues:

1. Reviewer: I would advise combining descriptive statistics into one table, and including one further table with the results of the ANOVA

Response: Thank you for this suggestion. I have combined descriptive statistics (previously Tables 1 and 2) into one table (now Table 2). I also have added a table with ANOVA results (Table 3).

2. Reviewer: Title for tables should be revised to be more informative and according to APA rules

Response: Table titles have been updated and re-formatted accordingly.

3. Reviewer: Wording of the paper should be checked, it would be better to avoid sentences in the first person

Response: The paper is now in third person and wording has been carefully checked.

---

## [Decision Letter · Decision Letter 1]

27 Apr 2023

PONE-D-22-10305R1The doctor knows or the evidence shows: An online survey experiment testing the effects of source trust, pro-vaccine evidence, and dual-processing in expert messages recommending the child COVID-19 vaccination to parentsPLOS ONE

Dear Dr. Kikut,

Thank you for submitting your manuscript to PLOS ONE. After careful consideration, we feel that it has merit but does not fully meet PLOS ONE’s publication criteria as it currently stands. Therefore, we invite you to submit a revised version of the manuscript that addresses the points raised during the review process.

Please find Reviewer #3 comments below, containing several detailed points. I believe that you can swiftly address most of them to improve your manuscript. One of Reviewer #3 points, however, requires your special attention. It is related to the lack of preexisting vaccine beliefs measurement. Perhaps, in the light of that remark, it would be better to say about the effect of your IVs on vaccine beliefs, but not on persuasiveness, as you indeed do not evidence attitude changes. One more thing is how you formulate your hypotheses. I strongly suggest sticking to statistical - not causal - relationships. E.g., instead of saying "increase," it is better to say "positively related to." Statistical relationships are what you actually test in your study. Moreover, if you prefer to display your hypotheses in a separate section, clarify how they are based on your considerations in the preceding section. It will improve your communication, especially among readers who will not necessarily read your entire text.

We look forward to receiving your revised manuscript.

Kind regards,

Wojciech Trzebinski, Ph.D.

Academic Editor

PLOS ONE

Journal Requirements:

Reviewers' comments:

Reviewer's Responses to Questions

**Comments to the Author**

1. If the authors have adequately addressed your comments raised in a previous round of review and you feel that this manuscript is now acceptable for publication, you may indicate that here to bypass the “Comments to the Author” section, enter your conflict of interest statement in the “Confidential to Editor” section, and submit your "Accept" recommendation.

Reviewer #3: (No Response)

Reviewer #4: All comments have been addressed

2. Is the manuscript technically sound, and do the data support the conclusions?

Reviewer #3: Partly

Reviewer #4: Yes

3. Has the statistical analysis been performed appropriately and rigorously? 

Reviewer #3: Yes

Reviewer #4: Yes

4. Have the authors made all data underlying the findings in their manuscript fully available?

Reviewer #3: No

Reviewer #4: No

5. Is the manuscript presented in an intelligible fashion and written in standard English?

Reviewer #3: Yes

Reviewer #4: Yes

6. Review Comments to the Author

Reviewer #3: The author reports on a the effect of different message framing strategies on perceived message effectiveness and pro-vaccine beliefs. This is an important topic of interest to a variety of disciplines. While I am enthusiastic about the topic, opportunities for improvement are detailed below.

INTRODUCTION

Great context regarding the pandemic at the time of data collection.

1. A citation is needed for lines 51-53.

2. “Literature review” heading (line 59) should be changed to better reflect the content of each section being reviewed.

METHODS

3. Lines 169-171: Were there any group differences by period? Given the two phases, how did you ensure that participants did not complete the survey twice?

4. The exclusion of participants should be reported in the method section. The use of attention checks, or any other indicator of data quality (e.g., bot screening) should be reported. If these were not used, it should be noted as a limitation in the discussion section.

5. Given the lack of a priori power analysis, a sensitivity analysis should be conducted and reported in the method section.

6. Page starting on line 229: Because items for both measures were adapted from previous scales, specific information pertaining to how it was adapted (in text) and factor loadings (supplement is acceptable) should be reported.

7. Section beginning with Line 250: A more relevant subheading would be helpful. The following is unclear: 1) why are covariates not considered? Vaccination status of the child, age, and political affiliation are relevant to the outcomes based on the existing literature and warrant consideration. I appreciate that consideration of covariates was perhaps not pre-registered; however, one can certainly still conduct exploratory tests. And 2) randomizing order for assessing vaccine status does not control for social desirability bias in responding. The absence of measures pertaining to social desirability bias is a limitation to be briefly discussed.

RESULTS

8. An exploratory section testing covariates is warranted.

DISCUSSION

9. Lines 382-385: The study did not include an assessment of pre-message vaccine beliefs (which is a notable limitation). Thus, the claim that dual processing may play a role in shifting perceptions does not seem appropriate.

10. Childhood COVID-19 vaccine intention or uptake behavior are not assessed, and this is a limitation to be discussed given that the ultimate goal for health promotion messages is to increase uptake.

OTHER

11. The author notes no financial disclosure; however, the author also notes “budget constraints” impacting the timing of recruitment periods (lines 169-171). The funding source for participant payment should be reported.

12. Line 471 (Acknowledgements) – If Robert Hornik and Diana Mutz “provided expertise and support throughout all aspects of this study”, why are they not co-authors?

Reviewer #4: Interesting study looking at the relative contribution of trust and evidence to child COVID-19 vaccination beliefs. Substantive adjustments to the paper have been made, reviewer comments have been addressed in full and the manuscript is suitable for publication. Some minor writing errors still persist, but these can be identified through the final editorial stages.

7. PLOS authors have the option to publish the peer review history of their article (what does this mean?). If published, this will include your full peer review and any attached files.

Reviewer #3: No

Reviewer #4: No

---

## [Author Response · Author response to Decision Letter 1]

6 Jun 2023

Note: A properly formatted version of this response was uploaded as a word document with cover letter and is included in the manuscript file PDF. 

Editor 

Editor: Please find Reviewer #3 comments below, containing several detailed points. I believe that you can swiftly address most of them to improve your manuscript. One of Reviewer #3 points, however, requires your special attention. It is related to the lack of preexisting vaccine beliefs measurement. Perhaps, in the light of that remark, it would be better to say about the effect of your IVs on vaccine beliefs, but not on persuasiveness, as you indeed do not evidence attitude changes. 

Response: Thank you for this note. It is true this study does not measure dependent variables at baseline, and therefore does not explicitly measure change in dependent variables. Language suggesting measurement of change has been modified throughout. Specifically, the term “increase” was substituted with “influence” in the following sections:

Lines 28-30: Either highlighting evidence supporting vaccines or building trust in expert sources can increase influence parent vaccine support. Messages which include strong evidence supporting recommended behaviors may increase influence recommendation acceptance.

Line 223: A manipulation check measured whether trust cues increased influenced perceived source trustworthiness as intended.

Line 513: First, results show it is possible for public health messages to increase influence recommendation acceptance…

Editor: One more thing is how you formulate your hypotheses. I strongly suggest sticking to statistical - not causal - relationships. E.g., instead of saying "increase," it is better to say "positively related to." Statistical relationships are what you actually test in your study. 

Response: Thank you for this feedback. If relationships between independent and dependent variables were tested in an observational study, this would warrant caution against causal language. The experimental design of this study allows for a test of causal relationships with strong internal validity and causal language in the hypothesis is appropriate. However, given the feedback above, the wording of the hypotheses has been adjusted to remove language that suggests change was measured. The new hypotheses refer to positive “main effects” and negative “interactive effects” of independent variables on dependent variables. Changes to the language in lines 152-159 are shown below (new words underlined):

Old wording: 

Hypothesis 1: Building source trustworthiness will increase message effectiveness. 

Hypothesis 2: Providing evidence supporting vaccine safety and efficacy will increase message effectiveness.

Hypothesis 3: There will be an interactive effect of building source trustworthiness and providing evidence supporting vaccine safety and efficacy, such that each approach will more strongly increase message effectiveness when the other is not used.

New wording: 

Hypothesis 1: Building source trustworthiness will have positive main effects on message effectiveness. 

Hypothesis 2: Exposure to evidence supporting vaccine safety and efficacy will have positive main effects on message effectiveness.

Hypothesis 3: There will be a negative interactive effect of building source trustworthiness and providing evidence supporting vaccine safety and efficacy on message effectiveness, such that each approach will have stronger positive effects when the other is not used.

Editor: Moreover, if you prefer to display your hypotheses in a separate section, clarify how they are based on your considerations in the preceding section. It will improve your communication, especially among readers who will not necessarily read your entire text.

Response: The following underlined words were added to clarify that hypotheses were based on considerations in the preceding section (lines 150-151): 

Drawing from prior research and dual-processing theories, the following was hypothesized.

Reviewer #3

Reviewer: The author reports on a the effect of different message framing strategies on perceived message effectiveness and pro-vaccine beliefs. This is an important topic of interest to a variety of disciplines. While I am enthusiastic about the topic, opportunities for improvement are detailed below.

Response: Thank you for your thoughtful feedback. I hope the responses below address your concerns. 

INTRODUCTION

Reviewer: Great context regarding the pandemic at the time of data collection.

Response: Thank you.

1. Reviewer: A citation is needed for lines 51-53.

Response: Citations have been added to this section (now lines 53-55). 

2. Reviewer: “Literature review” heading (line 59) should be changed to better reflect the content of each section being reviewed.

Response: The literature review heading has been removed. Three sections of the introduction are now labeled: “Providing pro-vaccine evidence as a messaging approach” (line 61), “Building trust in the source as a messaging approach” (line 91), “Dual-processing and the potential interaction between approaches” (lines 104-105). 

METHODS

3. Reviewer: Lines 169-171: Were there any group differences by period? Given the two phases, how did you ensure that participants did not complete the survey twice?

Response: There were no demographic differences or differences between those recruited in December and those recruited in January. The Forthright Capabilities Panel allows researchers to exclude panelists who had completed a previous round of data collection. 

4. Reviewer: The exclusion of participants should be reported in the method section. The use of attention checks, or any other indicator of data quality (e.g., bot screening) should be reported. If these were not used, it should be noted as a limitation in the discussion section.

Response: All participants who completed the study were included. Forthright panelists are verified by Forthright. Therefore, bot screening was not necessary within the study. A sentence was added to the limitations section regarding lack of attention checks (lines 451-453): 

Third, attention checks were not included in this study. It is possible that limiting analyses to participants who passed attention checks would have increased likelihood of detecting effects on beliefs.

5. Reviewer: Given the lack of a priori power analysis, a sensitivity analysis should be conducted and reported in the method section.

Response: Thank you for this point. Power analyses have been conducted and a new section has been added to the method’s section (line 374-384):

Post-hoc sensitivity analysis

Following hypothesis tests, power levels were computed for each statistical anlysis using the sample size, mean outcome scores (PME and behavioral beliefs), and within-cell variances in the current study sample. The power to detect significant effects of treatments on PME exceeded 80% for main effects (trust: ß = 0.84; evidence: ß = 0.89) and was slightly below 80% for the interaction of trust and evidence (ß = 0.72). The power to detect significant effects of each treatment on behavioral beliefs was below 80% for both main and interaction effects (trust: ß= 0.64; evidence: ß= 0.27; interaction of trust and evidence: ß=0.39). This post-hoc analysis suggests the study’s sample size was appropriate for detecting treatment effects on PME but may have been too low to detect some significant effects on behavioral beliefs. 

6. Reviewer: Page starting on line 229: Because items for both measures were adapted from previous scales, specific information pertaining to how it was adapted (in text) and factor loadings (supplement is acceptable) should be reported.

Response: Wording modifications were made to the original scale to fit the population, behavior of interest, and experimental context. To preserve space in the manuscript, a table has been added to the supplement to detail wording modifications (see table below). The created scale summed the raw scores of the six items, and did not weigh by factor loadings. None the less, interitem reliability was strong (alpha=0.86, as noted in the manuscript). Details regarding results of principal components analysis are now included in the supplement (see table note below). 

The following underlined words were added to lines 255-257: 

Items were adapted from a previously validated scale (46) for the current study context. Supplementary materials detail modifications (S1B Table) and summarize results of principal components analysis for the created scale. 

A table was added to the supplement, with details of principal components analysis in the table note (underlined below):

S1B Table: Development of Perceived Message Effectiveness Scale from Zhao et al. (2011) 

[See properly formatted document attached for table]

Note. This table illustrates each item recommended in the perceived argument strength scale provided by Zhao et al. (2011) and the corresponding item adapted for the perceived message effectiveness (PME) scale used in the current study. As described in the final column of the table, modifications were made to fit the population, behavior of interest, and experimental context. Principal components analysis on the modified scale showed the eigenvalue for the first component (“This quote will be effective in encouraging parents to get their children vaccinated against COVID-19”) was 3.75 and no other component had an eigenvalue meaningfully higher than 1. All other components loaded positively on the first component. Eigenvector loadings were 0.458, 0.464, 0.210, 0.457, 0.474, and 0.314. 

7. Reviewer: Section beginning with Line 250: A more relevant subheading would be helpful. The following is unclear: 1) why are covariates not considered? Vaccination status of the child, age, and political affiliation are relevant to the outcomes based on the existing literature and warrant consideration. I appreciate that consideration of covariates was perhaps not pre-registered; however, one can certainly still conduct exploratory tests. And 2) randomizing order for assessing vaccine status does not control for social desirability bias in responding. The absence of measures pertaining to social desirability bias is a limitation to be briefly discussed.

Response: The “Additional Measures” Section now includes two relevant sub-headers, “Decision-maker” and “Child vaccination status”. In response to point 1, political affiliation, age, and other demographic variables should be the same across conditions due to randomization and should not influence differences between treatment and control conditions. I appreciate the suggestion to consider contingency effects, but do not have a large enough sample size to add additional interactions with sufficient power. Such an analysis would distract from the purpose and findings of this study. 

In response to point 2, I have included a sentence to the discussion mentioning that social desirability bias may impact responses (lines 453-459):

Finally, it is worth noting that all conditions, including the control, conveyed information supporting child vaccination. Therefore, responses may have been influenced by social desirability bias. Future work should control for social desirability.

RESULTS

8. Reviewer: An exploratory section testing covariates is warranted.

Response: See response to point 7 above. Controlling for potential covariates should not influence results, due to randomization. The sample size is not adequate to test for contingency effects with sufficient power. 

DISCUSSION

9. Reviewer: Lines 382-385: The study did not include an assessment of pre-message vaccine beliefs (which is a notable limitation). Thus, the claim that dual processing may play a role in shifting perceptions does not seem appropriate.

Response: Thank you. The assumption of the experimental design is that the control condition represents the population absent treatment, and that the difference between the control and treatment conditions captures treatment effects. However, it is true that dependent variables were not measured at baseline. Further, the intention of this paragraph was to convey that the topic of treatment messages was real and likely familiar to participants at the time of the study, in contrast to hypothetical topics often used to test theory. To better illustrate this point, lines 414-416 have been updated as follows: 

Old sentence: Critics of dual-processing theories have argued that the model is only useful for explaining attitude formation, rather than attitude change 

New sentence: Critics of dual-processing theories have argued that the model is only useful for explaining attitude formation in hypothetical scenarios, rather than attitude formation in the context of real health recommendations 

Lines 419-422 have been updated as follows: 

Old sentence: The present study’s finding, that pro-vaccine evidence and source trustworthiness each had a stronger effect on parent vaccine endorsement when the other was absent, suggests dual-processing may indeed play a role in shifting perceptions.

New sentence: The present study’s finding, that pro-vaccine evidence and source trustworthiness each had a stronger effect on parent vaccine endorsement when the other was absent, suggests dual-processing may indeed play a role in influencing perceptions about real-world guidelines.

10. Reviewer: Childhood COVID-19 vaccine intention or uptake behavior are not assessed, and this is a limitation to be discussed given that the ultimate goal for health promotion messages is to increase uptake.

Response: The following paragraph was added to address the limitation (lines 460-468):

Despite these limitations, the detection of significant main effects on behavioral beliefs and main and significant interaction effects on PME should be considered strong indicators of the potential influence of both trust and exposure on parent vaccine decisions. Prior literature has shown behavioral beliefs strongly predict subsequent behavior for a range of contexts (56), including COVID-19 vaccination (57,58). PME has also been validated as a strong predictor of behavioral beliefs, intention, and behavior (44,45). Therefore, though this study did not measure behavior change, treatment effects on beliefs and PME—both antecedents of behavior—are promising indicators of influence on subsequent child vaccination decisions. 

OTHER

11. Reviewer: The author notes no financial disclosure; however, the author also notes “budget constraints” impacting the timing of recruitment periods (lines 169-171). The funding source for participant payment should be reported.

Response: Funds were made available through an internal department grant. There is no external funding to report. 

12. Reviewer: Line 471 (Acknowledgements) – If Robert Hornik and Diana Mutz “provided expertise and support throughout all aspects of this study”, why are they not co-authors?

Response: I designed the study and conducted data collection, analysis, interpretation, and writing independently. At the time of this study, I was a PhD student. Along the way, I discussed the study with my advisor (Bob Hornik), but he was not involved to the extent of a co-author. Diana Mutz taught the class for which this paper was initially written and offered expertise and support as an instructor for this course. It was not seen as appropriate by to include either as a co-author. 

Reviewer #4

Reviewer: Interesting study looking at the relative contribution of trust and evidence to child COVID-19 vaccination beliefs. Substantive adjustments to the paper have been made, reviewer comments have been addressed in full and the manuscript is suitable for publication. Some minor writing errors still persist, but these can be identified through the final editorial stages.

 Response: Thank you!

---

## [Decision Letter · Decision Letter 2]

26 Jun 2023

The doctor knows or the evidence shows: An online survey experiment testing the effects of source trust, pro-vaccine evidence, and dual-processing in expert messages recommending child COVID-19 vaccination to parents

PONE-D-22-10305R2

Dear Dr. Kikut,

We’re pleased to inform you that your manuscript has been judged scientifically suitable for publication and will be formally accepted for publication once it meets all outstanding technical requirements.

Kind regards,

Wojciech Trzebiński, Ph.D.

Academic Editor

PLOS ONE

Additional Editor Comments (optional):

While completing the required amendments, please fix the following two minor issues:

(1) make sure you use "interaction effect" consistently throughout the paper (in some places, you use "interactive effects")

(2) make sure you use "pro-vaccination belief" when you refer to this measurement (in some places, you use just "belief," which may be unclear for readers).

Reviewers' comments:

Reviewer's Responses to Questions

**Comments to the Author**

1. If the authors have adequately addressed your comments raised in a previous round of review and you feel that this manuscript is now acceptable for publication, you may indicate that here to bypass the “Comments to the Author” section, enter your conflict of interest statement in the “Confidential to Editor” section, and submit your "Accept" recommendation.

Reviewer #3: All comments have been addressed

2. Is the manuscript technically sound, and do the data support the conclusions?

Reviewer #3: Yes

3. Has the statistical analysis been performed appropriately and rigorously? 

Reviewer #3: Yes

4. Have the authors made all data underlying the findings in their manuscript fully available?

Reviewer #3: No

5. Is the manuscript presented in an intelligible fashion and written in standard English?

Reviewer #3: Yes

6. Review Comments to the Author

Reviewer #3: The concerns have been adequately addressed. Thank you.

7. PLOS authors have the option to publish the peer review history of their article (what does this mean?). If published, this will include your full peer review and any attached files.

Reviewer #3: No

---

## [Editor Report · Acceptance letter]

14 Jul 2023

PONE-D-22-10305R2 

The doctor knows or the evidence shows: An online survey experiment testing the effects of source trust, pro-vaccine evidence, and dual-processing in expert messages recommending child COVID-19 vaccination to parents 

Dear Dr. Kikut:

I'm pleased to inform you that your manuscript has been deemed suitable for publication in PLOS ONE. Congratulations! Your manuscript is now with our production department. 

Kind regards, 

on behalf of

Dr. Wojciech Trzebiński 

Academic Editor

PLOS ONE